# An Organismal Perspective on the Warburg Effect and Models for Proliferation Studies

**DOI:** 10.3390/biology12040502

**Published:** 2023-03-26

**Authors:** Neil W. Blackstone, Weam S. El Rahmany

**Affiliations:** Department of Biological Sciences, Northern Illinois University, DeKalb, IL 60115, USA

**Keywords:** aerobic glycolysis, anaerobic respiration, cancer, cnidarians, hydroids, lactate, mitochondria, proliferation, reactive oxygen species, succinate

## Abstract

**Simple Summary:**

Considerable interest in the physiology of proliferation has been generated by human cancers. Cancers often share features with other proliferative cells that are exemplified by the Warburg effect, which involves an altered metabolism with less oxygen uptake and greater lactate secretion. While some of the Warburg effect can be rationalized as production of biosynthetic precursors to fuel proliferation, lactate secretion does not fit this paradigm as it wastes these valuable precursors. In this context, examining proliferation in organisms that are capable of anaerobic pathways other than lactate production can be illuminating. Many animal species, and indeed eukaryotes in general, can carry out other bioenergetic reactions that do not use oxygen. These reactions frequently involve mitochondria and employ such seemingly baroque processes as running the Krebs cycle, the metabolic heart of the cell, backwards. Detailed study of these processes may require development of new animal model systems. For example, colonial marine hydroids frequently exhibit a stage in the life cycle that only undergoes cellular proliferation and never produces gametes. In contrast, traditional animal models such as worms, flies, and mice undergo only limited proliferation before gamete production. The crucial unresolved issues related to the Warburg effect could thus be understood.

**Abstract:**

Interest in the physiology of proliferation has been generated by human proliferative diseases, i.e., cancers. A vast literature exists on the Warburg effect, which is characterized by aerobic glycolysis, diminished oxygen uptake, and lactate secretion. While these features could be rationalized via the production of biosynthetic precursors, lactate secretion does not fit this paradigm, as it wastes precursors. Forming lactate from pyruvate allows for reoxidizing cytosolic NADH, which is crucial for continued glycolysis and may allow for maintaining large pools of metabolic intermediates. Alternatively, lactate production may not be adaptive, but rather reflect metabolic constraints. A broader sampling of the physiology of proliferation, particularly in organisms that could reoxidize NADH using other pathways, may be necessary to understand the Warburg effect. The best-studied metazoans (e.g., worms, flies, and mice) may not be suitable, as they undergo limited proliferation before initiating meiosis. In contrast, some metazoans (e.g., colonial marine hydrozoans) exhibit a stage in the life cycle (the polyp stage) that only undergoes mitotic proliferation and never carries out meiosis (the medusa stage performs this). Such organisms are prime candidates for general studies of proliferation in multicellular organisms and could at least complement the short-generation models of modern biology.

## 1. Introduction

Considerable interest in the physiology of proliferation has been generated by human proliferative diseases, i.e., cancers. A vast literature exists on the so-called Warburg effect [1,2,3,4,5,6,7,8,9], which is characterized by aerobic glycolysis, diminished oxygen uptake, and lactate secretion [9]. While the Warburg effect could partly be rationalized as a way of producing biosynthetic precursors for proliferation, lactate excretion does not fit this paradigm, as it wastes these valuable precursors.

Pyruvate may be reduced to lactate to reoxidize cytosolic NADH and allow for glycolysis (which must reduce NAD^+^) to continue. Apart from some vertebrates, very few metazoans primarily use this pathway to maintain redox balance in the absence of molecular oxygen [10,11,12]. More typically, under long-term anaerobic conditions, cytosolic NADH is reoxidized by, for example, forming malate from oxaloacetate. Malate could then be processed in mitochondria, with the additional NADH reoxidized using a truncated electron transport chain running the Krebs cycle backwards and forming succinate, which may then be converted into products such as acetate and propionate before excretion. As with other eukaryotes, metazoans do not to operate the entire Krebs cycle in reverse, although many prokaryotes do [13].

Remarkably, the study of proliferation is largely confined to metazoans with a limited cellular physiology. What the Warburg effect is and, more generally, what the physiology of proliferation looks like in organisms that have broader anaerobic capabilities, at least for eukaryotes, are unaddressed. Is lactate still produced or does lactate production reflect a fundamental metabolic constraint of some metazoans? While some aspects of proliferation are dependent on the availability of biosynthetic precursors, other aspects (e.g., protein translation) depend on large amounts of ATP. During proliferation, why is pyruvate not processed in the Krebs cycle and electron transport chain to maximize ATP formation?

Further, the best-studied metazoans (e.g., worms, flies, and mice) undergo limited proliferation before initiating the sexual phase of the life cycle. In contrast, some metazoans (e.g., colonial marine hydrozoans with a metagenic life cycle) exhibit a stage in the life cycle (the polyp stage) that only undergoes mitotic proliferation and never carries out meiosis to form gametes (this is conducted in the medusa stage). The proliferative polyp stage can be extraordinarily long-lived (e.g., we have had clones in culture for >30 years). Such organisms are prime candidates for general studies of proliferation in multicellular organisms and could minimally complement the short-generation models of modern biology.

Two robust fields of intellectual inquiry have existed independently for some time. On the one hand, the Warburg effect is studied by biomedical scientists using human cells. On the other hand, organismal biologists and physiologists investigate the anaerobic metabolism using a host of nonmodel organisms. We attempt to unite these distinct fields of inquiry here. It is not our goal to extend the boundaries of either field or to elaborate their rich details. Both fields have developed large literatures and have great numbers of active researchers. Our goal is to connect these fields for useful synergisms.

Further, to broaden the study of proliferation, we outline the biological features that should be included in these more general studies. First, it is essential to examine how the Warburg effect manifests in organisms that are capable of a sophisticated (for eukaryotes) anaerobic metabolism. What are the metabolic features of proliferative cells and organisms that are capable of reoxidizing NADH using pathways other than lactate production or the canonical electron transport chain? Second, the model systems of modern biology were mostly developed to study genetics. These model organisms rapidly complete the proliferative phase of the life cycle and may be largely postmitotic as adults. We suggest including organisms with life cycle stages that are entirely mitotic and long-lived. Cancer cells proliferate mitotically, but they never undergo meiosis. Hence, it is not unreasonable to develop models of proliferation in general and cancer in particular that have the same characteristics. Lastly, keeping in mind the admonishment from Martin et al. [10] (p. 5): “the stories that can be read from, and into, genome sequences can differ substantially from the actual physiology of the organisms”, we recognize that, while genomics can be a useful tool, it is no substitute for detailed physiological studies.

This research program provides answers to various questions related to the Warburg effect that have remained unresolved for nearly a century, including the following:In Metazoa, how generally do proliferative cells and organisms exhibit the Warburg effect?Do high rates of proliferation result in lactate formation or are alternative pathways used to reoxidize NADH?Why is NADH not reoxidized using the entire electron transport chain with oxygen as a terminal electron acceptor?

## 2. A Closer Look at Anaerobic Metabolism in Eukaryotes

### 2.1. Eukaryotic Metabolism

In terms of structure, function, life cycle, genes, and genomes, eukaryotes are remarkably complex. On the other hand, in contrast to prokaryotes, both aerobic and anaerobic metabolism is highly stereotyped in eukaryotes [10,11,12]. Aerobic fermentation (the Crabtree effect) is well-known in certain fungi [10]. When subject to transitory hypoxia, metazoans, particularly those inhabiting the intertidal, form opines [10,14,15]. Analogous to lactate formation, opines are formed from the condensation of pyruvate and an amino acid, and this process reoxidizes NADH. Opines are typically stored until normoxic conditions allow for their oxidation in mitochondria. Hence, opines signify transitory hypoxia [10] and are not a major end product in a continuously proliferative system.

Under more long-lasting anaerobic conditions, many metazoans [10,11,12] carry out malate dismutation in which oxaloacetate is converted into malate to reoxidize cytosolic NADH using the enzyme malate dehydrogenase. In mitochondria, some of the malate is subsequently oxidized and some is reduced [10]. While technically not respiration, it can be thought of in this way since mitochondrial NADH is reoxidized by converting malate into fumarate and then succinate, i.e., running the Krebs cycle backwards while using a portion of the electron transport chain. Crucially, this involves using rhodoquinone, not ubiquinone, as a mobile electron carrier [10]. The formed end products include acetate, propionate, and succinate (Figure 1).

Martin et al. [10] provide a detailed examination of a number of metazoans that exhibit malate dismutation, e.g., flatworms, roundworms, mollusks, and annelids. While individual taxa exhibit some idiosyncratic variation, the general themes of malate dismutation remain constant. Rather than being converted into pyruvate, phosphoenolpyruvate is converted into oxaloacetate in a reaction catalyzed by ATP-dependent phosphoenolpyruvate carboxykinase, which takes up carbon dioxide. The NADH from glycolysis is then reoxidized by converting oxaloacetate into malate, which is then imported into mitochondria.

In anaerobic mitochondria, malate dismutation uses rhodoquinone (RQ) due to its strong electron donor potential relative to ubiquinone (UQ). The alternative splicing of a specific exon provides the capability to synthesize RQ [16]. Two critical residues distinguish this exon. Three additional genes associated with an RQ-based anaerobic metabolism were also identified [17]. As described in more detail below, bioinformatic methods could, thus, be used to search for the signature of anaerobic mitochondria in available genomic data prior to undertaking the necessary experiments to investigate the actual physiology.

### 2.2. Warburg Effect

The modern conceptualization of the Warburg effect focuses on aerobic glycolysis and lactate secretion [9]. The former was rationalized in terms of the need for biosynthetic precursors rather than ATP when proliferating; the latter, however, wastes potential precursors. Thus, as indicated by DeBerardinis and Chandel [9], “the truth is that the reason why many proliferating cells display the Warburg effect is still not fully understood”. At the most basic level, forming lactate from pyruvate allows for reoxidizing cytosolic NADH, which is crucial for continued glycolysis since glycolysis reduces NAD^+^. Lunt and Vander Heiden [4] indicated that the Warburg effect allows for maintaining large pools of glycolytic intermediates that are then readily available as biosynthetic precursors. Lunt and Vander Heiden, and indeed much of the biomedical literature, have conceptualized the Warburg effect as adaptive in some fashion.

Alternatively, lactate production may not be adaptive at all, but rather may reflect the fundamental metabolic constraints of eukaryotes. While some aspects of proliferation (e.g., synthesizing proteins) require large amounts of ATP, overall cellular proliferation may have greater requirements for biosynthetic precursors. Many of these precursors form from glycolytic and Krebs cycle intermediates. As with any chemical reaction, an excess of product slows down or stops substrate processing. Hence, metabolizing pyruvate in the Krebs cycle and electron transport chain may produce excess ATP that may then inhibit the Krebs cycle, glycolysis, and the production of biosynthetic precursors [18]. Similarly, proliferative cells may monopolize nutrients to fuel their growth. Excessive amounts of nutrients may simply overwhelm the ability of cells to process them completely. For instance, the mitochondrial NADH shuttles that allow for reoxidizing cytosolic NADH may simply be overloaded in proliferative cells, so NADH must be reoxidized with conversion into lactate to maintain redox balance [19]. Excess amounts of ATP and NADH may result in a related constraint. Abundant NADH feeds electrons into the mitochondrial electron transport chain. At the same time, end-product inhibition resulting from an excess of ATP in relation to metabolic demand slows the movement of electrons through this chain. Consequently, these electrons may be cast off on molecular oxygen, leading to the formation of harmful levels of reactive oxygen species (ROS) [20].

These hypotheses are not mutually exclusive and likely operate together. The finding that proliferative cells use, for example, malate dismutation would have implications in this context. With malate dismutation, cytosolic NADH is reoxidized by the formation of malate from oxaloacetate (Figure 1). Malate then enters mitochondria, although the exact mechanism remains unclear from the available literature. While importing malate into mitochondria avoids one of the NADH shuttles (the glycerol 3-phosphate shuttle), it may involve the other (the malate–aspartate shuttle). The variety of mitochondrial transporters may not be fully characterized in nonmodel metazoans; thus, the possibility remains that mitochondria could be used in malate dismutation without using NADH shuttles and with greater ATP yield than that of lactate formation [10]. An absence of lactate formation would imply that there is no special role for lactate in cellular proliferation. Further, malate dismutation may imply some advantage for greater ATP yields if overloaded NADH shuttles can be circumvented. While malate dismutation results in a greater ATP yield than that of lactate formation, this yield is still considerably smaller than that produced when oxygen is the terminal electron acceptor. Malate dismutation may, thus, represent an effective compromise: circumventing overloaded NADH shuttles and producing more (roughly 2.5 times as much [10]) ATP than that of lactate formation while avoiding levels of ATP that shut down the Krebs cycle and glycolysis, and result in excessive ROS formation.

The physiology of proliferation in metazoans that could reoxidize NADH via pathways other than the canonical electron transport chain or converting pyruvate into lactate [10,11,12] may, thus, provide crucial data to understanding the Warburg effect. As indicated above, such pathways may involve malate dismutation and form succinate, among other products. In this pathway, the malate dehydrogenase enzyme reoxidizes cytoplasmic NADH by forming malate (Figure 1), and mitochondrial NADH shuttles may not be required. Malate can be converted into succinate as described above. In the presence of molecular oxygen, however, succinate can lead to reverse electron transfer and high levels of ROS [21], which may explain why succinate is typically converted into other products. If a proliferating organism that is capable of other anaerobic pathways nevertheless primarily produces lactate, this strongly suggests an undiscovered role of lactate during proliferation, e.g., in signaling.

### 2.3. One Ring to Rule Them All

Life generally consists of two aspects: (1) extracting energy from the environment and converting it into useful forms and (2) using some of that energy to replicate a store of information [22,23,24]. Unsurprisingly, disciplinary boundaries reflect this dichotomy, e.g., biochemistry and molecular biology. In this context, the late 20th century neglect of intermediary metabolism in favor of informational molecules, i.e., genes and genomes, is understandable. Nevertheless, as Lane [18] (p. 6) suggested, “…textbook biochemistry is simultaneously galvanizing new paradigms on the origin of life and cancer…”. At the center of it all lies the Krebs cycle—the one ring that rules them all. This is perhaps the central lesson of the Warburg effect: metabolism matters. How cells process substrate could impact a variety of related processes. Again, as suggested by Lane [18] (p. 20): “…molecules accumulating in the Krebs cycle can signal the state of the cell to the genes, switching on or off hundreds or even thousands of genes. Far from being dusty textbook chemistry, we now know that different patterns of metabolic flux through the Krebs cycle can generate powerful, if ambiguous, signals”. Appropriately, since Krebs and Warburg’s careers intertwined [25], the Krebs cycle and the Warburg effect intertwine as well.

## 3. New Model Organisms

In a broad outline, the early history of metazoans can be characterized by a dichotomy between clonal and colonial animals (e.g., placozoans, sponges, cnidarians), and stem bilaterians. From this view, stem ctenophores were also clonal. In the context of this oversimplified but still useful dichotomy, stem bilaterians were described as “assembly-line animals” that grow rapidly, disperse widely, and seek out patches of food to consume before sexually reproducing [26]. Clonal and colonial metazoans, on the other hand, have proliferated continuously, and responded to environmental signals to temper and guide this proliferation [27]. This dichotomy is reflected in the model systems of modern biology, which largely focus on the former and exclude the latter [28].

### 3.1. Model Systems for Genetics

In the early 20th century, a number of prominent scientists focused on the problem of regeneration, using clonal or colonial animals as their models [29]. After 1910, however, Thomas Hunt Morgan devoted his studies solely to transmission genetics using *Drosophila*. Of course, *Drosophila* is the archetypal assembly-line animal, virtually postmitotic as an adult and incapable of reproduction except via gametes. Morgan’s “fly room” became legendary, and much of the early history of the field of genetics was closely tied to *Drosophila*.

### 3.2. Model Systems for Proliferation

Ideally, studies of proliferation should focus on a life-cycle stage of an organism that is capable of unlimited proliferation, at least on the basis of human time scales. Clones of colonial marine hydroids such as *Hydractinia symbiolongicarpus*, *Podocoryna carnea,* and *Eirene* sp. have been maintained in our laboratory for decades. Periodically, with the exception of *H. symbiolongicarpus*, these clones produce medusae that reproduce sexually [30]. Meanwhile, the polyp stages of these species proliferate mitotically and, in the presence of abundant nutrients, exhibit exuberant patterns of growth (Figure 2). Proliferation can be regulated simply by manipulating the supply of nutrients.

### 3.3. Anaerobic Metabolism in Cnidarians

In many ways, cnidarians, which represent an early-diverging group of metazoans (Figure 3), remain poorly known. While there is evidence of anaerobic metabolism in several cnidarian groups [14,15,31], Martin et al. [10] represented the cnidarians as exclusively possessing obligately aerobic mitochondria (i.e., those that function only in the presence of oxygen). This is not necessarily a contradiction; anaerobic metabolism is not equivalent to facultatively anaerobic mitochondria (i.e., those that function both in the presence of oxygen and in its absence). Under anaerobic conditions, lactate or opine production does not involve mitochondria [10]. Nevertheless, Livingstone [14] listed succinate as a product of cnidarian anaerobic metabolism. As described above, under anaerobic conditions, succinate can be produced by mitochondria by running the Krebs cycle backwards and utilizing a portion of the electron transport chain [10,11,12,14]. To better resolve the character states of cnidarians and thus clarify the course of metabolic evolution, further data are needed.

While physiological studies are necessary, some insight can be gained from genetic data. A key difference in the function of the aerobic compared to the anaerobic mitochondrial electron transport chain is that the former uses ubiquinone (UQ) as an electron carrier, while the latter used rhodoquinone (RQ) (Figure 4). In the anaerobic metabolism, RQ is favored because of its strong electron donor potential relative to UQ. The alternative splicing of a specific exon of COQ-2e provides the capability to synthesize RQ [16]. Two critical residues distinguish this exon.

Besides RQ itself, anaerobic mitochondria require several other genetic variants compared to aerobic mitochondria. In particular, variants of three additional genes are associated with an RQ-based anaerobic metabolism [17]. The best-characterized are those associated with the quinone-binding pocket of succinate dehydrogenase (Complex II) [17]. Under aerobic conditions, Complex II acts as a dehydrogenase and converts succinate into fumarate, transferring two electrons onto UQ as the entry point into the mitochondrial electron transport chain (ETC). Under anaerobic conditions, on the other hand, this reaction is reversed (Figure 4). Complex II then acts as a fumarate reductase, and the two electrons required to convert fumarate into succinate must come from RQ. Under these conditions, Complex II acts as a point for electrons to exit a truncated ETC, and fumarate, not oxygen, is the terminal electron acceptor. Thus, aerobic and anaerobic duties require Complex II to use both UQ and RQ, and the quinone-binding pocket of Complex II must be able to dock both UQ and RQ. In purely aerobic mitochondria, Complex II only binds UQ. UQ and RQ are highly similar in structure, but differ at Position 2 of their benzoquinone rings. UQ has a methoxy group, whereas RQ has an amine group [17]. Complex II in mitochondria with anaerobic capabilities must be able to bind both ring structures, and this difference requires a variant amino acid surrounding the quinone-binding pocket of Complex II and the related nucleotide sequence changes. A single residue in the quinone-binding pocket of Complex II, subunit MEV-1, is required for RQ utilization [33].

Besides COQ-2e and Complex II, there are two other important indicators of an RQ-based metabolism [17]. TDO-2 acts in the RQ synthesis pathway, while ETFDH is a direct docking site for RQ, facilitating the transfer electrons out of the ETC and onto terminal acceptors such as fumarate. These four enzymes are required for RQ metabolism in *C. elegans* [16,17,33]. Comparing facultative anaerobic mitochondria to those that are purely aerobic, all four proteins exhibit specific changes in critical regions.

## 4. Do Cnidarians Have Anaerobic Mitochondria? A Bioinformatic Approach

### Anaerobic Metabolism in Cnidarians

Using four required and specific proteins for RQ metabolism [16,17,33], tblastn searches were conducted for available cnidarian genomes. As shown in Table 1, some cnidarian models exhibit the character states of aerobic mitochondria, while some exhibit anaerobic character states. Generally, sea anemones and hydroids comprise the latter. As discussed above, four proteins were examined. The first, COQ-2 (accession number for *Caenorhabditis elegans*, NP_871684.1), exhibited two critical residues (L204, S243) that are required for RQ-based anaerobic metabolism [16]. These two residues sit very close to the substrate in the active site of the COQ-2 enzyme. As shown in Table 1, *Acropora* and *Exaiptasia* lack both residues, while *Nematostella* exhibits both, and *Hydra* exhibits one with a variant residue at the other position. Other corals generally exhibit aerobic character states. Anaerobic mitochondria in *Nematostella* may be highly beneficial, since it is found in the intertidal and shallow subtidal. Various species of *Hydractinia*, *Podocoryna,* and *Eirene* are also intertidal and shallow subtidal. These results suggest that *Acropora*, studied by Linsmayer et al. [15], has obligately aerobic mitochondria and could hence only produce opines when anaerobic [15]. On the other hand, the anaerobic metabolism in some cnidarians likely extends beyond merely opine production.

The second protein associated with RQ-based metabolism is MEV-I, a Complex II subunit (accession number for *Caenorhabditis elegans*, NP_001366681.1). A single residue diagnoses the form used in anaerobic metabolism [33]. This residue, located near the rhodoquinone ring’s binding site [33], is found in *Hydra*, *Nematostella*, and other sea anemones, and one of the two surveyed soft corals. Hard corals (hexacorals), however, generally lack this residue. TDO-2 is the third protein required for RQ-based anaerobic metabolism (accession number for *Caenorhabditis elegans*, NP_498284.1) that exhibits one critical residue, P133, which is part of the PLD loop required for TDO-2 enzyme activity [34] and RQ-metabolism [17]. Most examined cnidarians exhibited this residue. Lastly, ETFDH (accession number for *Caenorhabditis elegans*, NP_001379625.1) is also required for RQ-based metabolism [17]. A critical residue (F437) for the anaerobic form is near the quinone binding site [35]. Remarkably, all cnidarians exhibit this critical residue, suggesting the presence of anaerobic mitochondria, if not currently then at least in a common ancestor.

## 5. Experimental Approach to Clarifying the Warburg Effect

### 5.1. Model Choice 

Two prominent cnidarian models, *Hydra* and *Nematostella*, either have facultative anaerobic mitochondria or have had them in a recent ancestor (Table 1). While these species are distinct phylogenetically and ecologically, namely, a freshwater hydra and a shallow-water marine actiniarian anemone, they are both solitary, noncolonial polyps that lack a medusa stage. For the most part, these polyps can reproduce via either sexual or asexual means, although there are strains or populations of both that favor one or the other mode of reproduction. Nevertheless, there is no canonical separation between the asexual and sexual stages in the life cycle. Such a separation is found in hydroids that exhibit a metagenic life cycle. In such hydroids, the nutrient manipulation of the polyp-stage replicates of the same genetic clone could produce rapidly or slowly proliferating treatments without the interference of sexual reproduction. We have been monitoring such treatments for several years for a clone of *Eirene* sp. [36] and, more recently, for *Podocoryna carnea* (Figure 2).

### 5.2. Expected Results

When subject to nutrient manipulation, differences in growth rate and form were expected and found [37]. Perhaps remarkably, differences in oxygen uptake were also found [36]. When adjusted for total protein, the colonies of the nutrient-scarce treatment showed greater oxygen uptake than that of the colonies of the nutrient-abundant treatment. This suggests that the rapidly proliferating nutrient-abundant colonies exhibit a Warburg-like metabolism. An obvious path forward would be to assess these species for RQ-based metabolism using genetic tools, and at the same time examine the metabolic end products of nutrient-abundant and -scarce colonies. Monitoring other physiological features (e.g., mitochondrial membrane potential and ROS) provides further relevant metabolic data.

## 6. Discussion

Warburg maintained that mitochondrial respiration was impaired in cancer cells [38]. While this is clearly not the case [39], the so-called Warburg effect has stimulated many studies and a large literature. This effect is characterized by aerobic glycolysis, diminished oxygen uptake, and lactate secretion [9]. While the Warburg effect can partly be rationalized as a way to produce the biosynthetic precursors for proliferation, lactate excretion does not fit this paradigm, as it wastes these valuable precursors. While not all cancers exhibit the Warburg effect, many noncancerous proliferative cells do. The Warburg effect may reveal crucial details of the physiology of proliferation. Nevertheless, lactate formation is characteristic of relatively few metazoans under anaerobic conditions. What does the Warburg effect look like in an organism that is capable of a more sophisticated anaerobic metabolism (for eukaryotes) than that of lactate formation? What does this suggest about the Warburg effect in general?

The models of modern biology (e.g., worms, flies, and mice) were never conceptualized as systems to study proliferation. Rather, they were prized for exactly the opposite features—a short period of proliferation followed by a rapid initiation of the sexual phase of the life cycle. These features rendered them valuable genetic models, but less than ideal systems to study proliferation. Indeed, worms (*C. elegans*) and flies (*Drosophila*) are essentially postmitotic as adults. On the other hand, biologists in the early 20th century had useful models to study proliferation, such as planarians and hydroids [28]. Those hydroids that have a metagenic life cycle might be particularly useful in this context, since they have a life-cycle (polyp) stage that only proliferates via mitosis and never forms gametes (the medusa stage performs this). Thus, metagenic hydroids that are capable of sophisticated anaerobic metabolism (e.g., malate dismutation [10]) might be the ideal model systems to examine in this context. Our sample of cnidarians was entirely deficient in terms of representatives of this group. While *Clytia hemisphaerica* has a genomic sequence, we were unable to find any sequences related to the four proteins described here using our methods.

In hydroids with a metagenic life cycle, the proliferation of the polyp state can be controlled simply by manipulating the supply of nutrients. Hence, an experimental approach involving nutrient manipulation is suggested. If highly proliferative (nutrient-abundant) hydroids create more products of anaerobic metabolism and consume less oxygen than less proliferative (nutrient scarce) hydroids do, the central tenets of the Warburg effect are supported in metazoans that are phylogenetically distant from vertebrates. On the other hand, if these products are not primarily lactate, a role for lactate in proliferation can be deemed unlikely. Rather, the reoxidization of NADH using anaerobic pathways in proliferating cells may reflect the fundamental constraints of eukaryotic physiology, resulting in the maladaptive excretion of end products that could be used as biosynthetic precursors and thus fuel proliferation. Three related hypotheses were mentioned above to explain this putative constraint: (1) the formation of ATP in quantities that trigger end-product inhibition, shutting down the Krebs cycle and glycolysis [18]; (2) an abundance of nutrients leading to overloaded mitochondrial NADH shuttles in rapidly proliferating cells and organisms [19]; (3) an abundance of nutrients leading to high ATP/ADP ratios, end-product inhibition, and highly reduced mitochondrial electron carriers, triggering high levels of ROS formation in rapidly proliferating cells and organisms [20]. These hypotheses are not mutually exclusive and almost certainly operate together. The proposed experiments here have implications regarding all three hypotheses. With regard to (1) and (2), mitochondria may be used in the anaerobic metabolism without using NADH shuttles and with greater ATP yield than that of lactate formation [10]. The absence of lactate or opine formation, and the corresponding presence of succinate or related products and mitochondrial membrane potential implies some advantage of greater ATP yields if overloaded NADH shuttles could be circumvented. Regarding (1) and (3), greater ROS formation in highly proliferative (nutrient-abundant) hydroids relative to less proliferative (nutrient-scarce) hydroids suggests that greater proliferation corresponds to greater risks from end-product inhibition and ROS associated with using oxygen as a terminal electron acceptor. Diverting substrate into other pathways may alleviate these risks. Indeed, preliminary data suggest that nutrient scarcity alleviates ROS levels, while nutrient abundance enhances them, perhaps requiring alternative metabolic solutions.

Since these are marine organisms with limited homeostatic capabilities, the experimental conditions are essentially those of seawater (pH 8.2, salinity roughly 30 parts per thousand, and high levels of carbonates, calcium, and magnesium) and ambient temperatures (roughly 20.5 °C). Given that these organisms are poorly known, there may be aspects of cellular metabolism besides those directly related to RQ-based metabolism that also show functional differences in facultative versus obligate aerobic mitochondria.

Available genomic data from cnidarian models show that all exhibit some features of facultative anaerobic mitochondria, but only one (*Nematostella*) exhibits all of the critical residues that are necessary for an RQ-based anaerobic metabolism. The simplest explanation for this pattern is that the common ancestor of cnidarians possessed facultative anaerobic mitochondria, while some modern species have lost these capabilities, but retain some of the enzymatic features, perhaps because of pleiotropy. Martin et al. [10] suggested that such a scenario may be general for eukaryotes. More complicated scenarios involving gains and losses are, of course, possible. Further sampling, particularly of marine hydroids, is necessary and allows for the formal reconstruction of ancestral character states [40]. Besides RQ-based characters, other aspects of malate dismutation may involve differences from human cell biology (e.g., the function of mitochondrial transporters).

## 7. Conclusions

The Warburg effect was conceptualized and examined within the confines of human biology, particularly that of proliferative diseases, i.e., cancers. An organismal perspective may help in clarifying the biological nature of this effect. Under anaerobic conditions, many metazoans exhibit metabolic capabilities that transcend using oxygen as a terminal electron acceptor or forming lactate. The question remains of what the Warburg effect looks like in these organisms. Further, the most common metazoan models (e.g., worms, flies, and mice) exhibit limited cellular proliferation before initiating the sexual phase of the life cycle. Such models are hardly suitable for studies of proliferation. Other metazoans are more favorable to such studies. Colonial marine hydroids with a metagenic life cycle were highlighted here. In these hydroids, the polyp stage of the life cycle only carries out mitotic proliferation. Such a dedicated stage of the life cycle allows for proliferation to be regulated entirely by nutrient manipulation. Nutrient-abundant (i.e., proliferative) and nutrient-scarce (less proliferative) treatments can be compared for physiological parameters related to the Warburg effect (e.g., production of succinate and other short-chain fatty acids, mitochondrial membrane potentials, and ROS). This perspective allows for a clearer picture of the Warburg effect to emerge.

## Figures and Tables

**Figure 1 biology-12-00502-f001:**
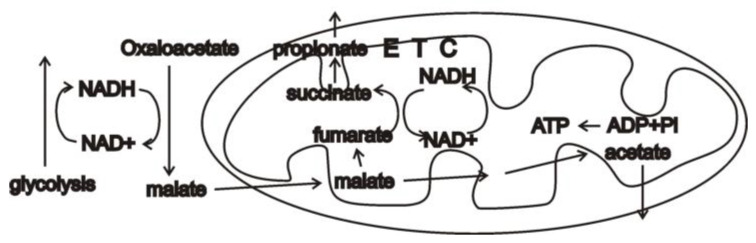
In the absence of oxygen, many metazoans carry out malate dismutation. Cytosolic NADH from glycolysis is reoxidized by converting oxaloacetate into malate. Malate is imported into mitochondria, where some of it is oxidized and some reduced. Thus-formed mitochondrial NADH powers a truncated electron transport chain (ETC) running the Krebs cycle backwards and using fumarate as the terminal electron acceptor. Excreted end products include acetate, propionate, and succinate.

**Figure 2 biology-12-00502-f002:**
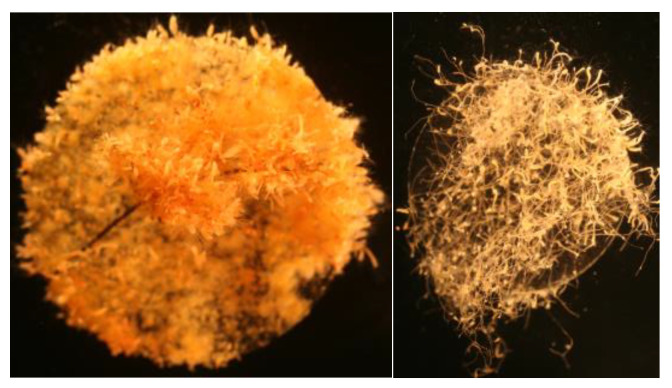
Colonies of polyps of *Podocoryna carnea* (left) and *Eirene* sp. growing on 18 mm diameter cover glass under nutrient-abundant conditions.

**Figure 3 biology-12-00502-f003:**
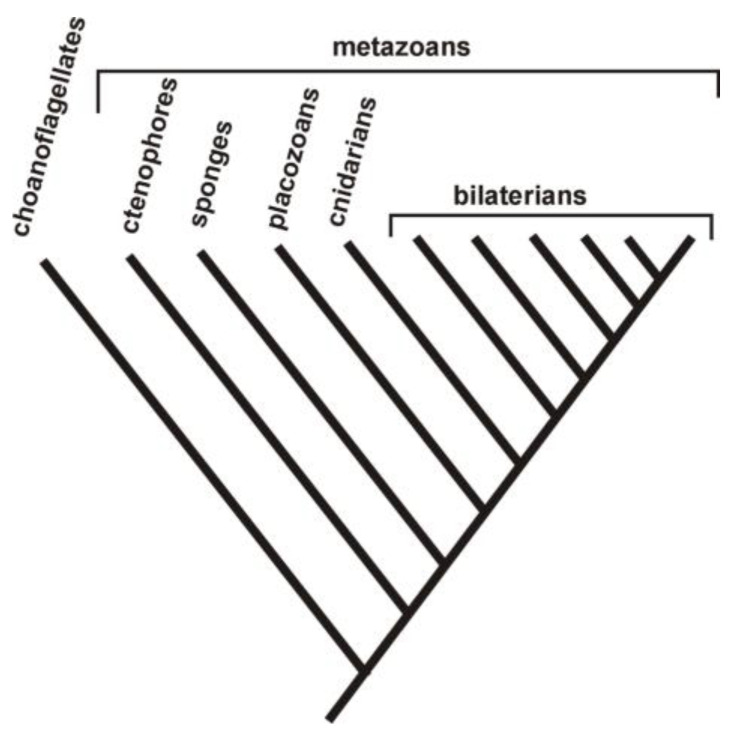
The five early-diverging groups of metazoans: ctenophores, sponges, placozoans, cnidarians, and bilaterians. The order of divergence is subject to considerable debate [32]. Choanoflagellates represent the sister group to Metazoa.

**Figure 4 biology-12-00502-f004:**
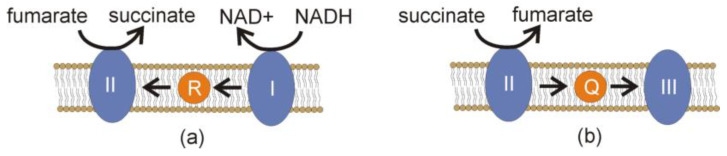
The use of ubiquinone and rhodoquinone in electron transport. (**a**) As part of the reductive branch of malate dismutation under anaerobic conditions, NADH is oxidized to NAD^+^ by complex I (I) of the electron transport chain. Electrons are carried by rhodoquinone (R) to fumarate reductase (II), reducing fumarate into succinate. (**b**) In the presence of oxygen, Complex II participates in the canonical electron transport chain, oxidizing succinate into fumarate and donating electrons to ubiquinone (Q), which carries the electrons to Complex III.

**Table 1 biology-12-00502-t001:** Proteins and their specific residues to RQ-based anaerobic metabolism for the available genetic data of cnidarians.

Species ^1^	COQ-2 Residues ^2^	MEV-1 Residue	TDO-2 Residue	ETFDH Residue
*Caenorhabditis elegans*	**L**204; **S**243	**G**71	**P**133	**F**437
*Homo sapiens*	F; A	I	A	C
*Hydra vulgaris*	**L**; A	**G**	**P**	**F**
*Nematostella vectensis* (sea anemone)	**L**; **S**	**G**	**P**	**F**
*Exaiptasia pallida* (sea anemone)	F; A	**G**	**P**	**F**
*Actinia tenebrosa* (sea anemone)	M; A	**G**	**P**	**F**
*Diadumene lineata* (sea anemone)	M; A	-	-	**F**
*Xenia* sp. Carnegie-2017 (soft coral)	I; A	I	A	**F**
*Dendronephthya gigantea* (soft coral)	F; A	**G**	S	**F**
*Orbicella faveolate* (hard coral)	F; A	I	**P**	**F**
*Stylophora pistillata* (hard coral)	F; A	I	**P**	**F**
*Acropora millepora* (hard coral)	F; A	I	**P**	**F**
*Acropora digitifera* (hard coral)	F; A	I	**P**	**F**
*Pocillopora damicornis* (hard coral)	F; A	-	**P**	**F**
* Blastomussa wellsi * (hard coral)	F; A	-	-	**F**
* Montipora capitata * (hard coral	F; A	-	-	**F**
* Galaxea fascicularis * (hard coral)	F; A	-	-	**F**
* Haliclystus octoradiatus * (stalked jellyfish)	M; A	-	-	**F**

^1^*C. elegans* exhibits anaerobic mitochondria, while *H. sapiens* does not. ^2^ Bold letters signify important residues for RQ-based anaerobic metabolism; amino acid abbreviations: A = alanine, C = cysteine, F = phenylalanine, G = glycine, I = isoleucine, L = leucine, M = methionine, P = proline, S = serine.

## Data Availability

No new data were created.

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
