# Peer review of "An Organismal Perspective on the Warburg Effect and Models for Proliferation Studies"

_biology, 2023, doi:10.3390/biology12040502_

Round 1

Reviewer 1 Report

The manuscript submitted by Blackstone & El Rahmany : “An organismal perspective on the Warburg effect” conveyed the impression of an almost philosophical point of view on the effect described by Otto Warburg in the context of tumor, on the perspective of other organism and a general view. Nevertheless, a clearer focus of this work on Cnidarian, for which the authors have an indisputable expertise, would have given this review a distinct direction.

Major Revision:

The author described a pathway and refers to it, which was described by different other authors e.g. | https://doi.org/10.1038/s41598-020-70202-y or https://doi.org/10.1016/j.bbabio.2020.148278.

The argumentation of these “other” authors is very similar:

Chetri et al.,: “During anaerobic metabolism, phosphoenolpyruvate (PEP) is converted to oxaloacetate via phosphoenolpyruvate carboxykinase (PEPCK), which is subsequently reduced by cMDH, reoxidizing the glycolytic NADH. This malate is transported to the mitochondria, where it is dismutated. Some malate is oxidized to acetate, while some are reduced to succinate and then metabolized to propionate.” https://doi.org/10.1038/s41598-020-70202-y

Salinas et al.,: “Glucose is converted into phosphoenolpyruvate by the glycolytic pathway, which is converted to oxaloacetate (not to pyruvate as in glycolysis), and then reduced to malate. Then, malate is imported into the mitochondrion, where a fraction is oxidized to acetate and CO2, and another fraction is converted to fumarate, which is then reduced to succinate by complex II.” “PEP is then converted to oxaloacetate (Oxac) by phosphoenolpyruvate carboxy-kinase” https://doi.org/10.1016/j.bbabio.2020.148278.

However, these interpretations of this pathway is the scientific basic for the argumentation of the manuscript submitted by Blackstone & El Rahmany .

I do not understand this pathway described by “other” authors (e.g. Salinas et al. or Chetri et al.), because in a biochemical sense phosphoenolpyruvate (PEP) cannot be converted into oxaloacetate. The reaction of a C3 into a C4-body is a carboxylation, not a convertion. Thus, carboxylation is not catalyzed by PEP, as stated by these “other” authors, because the phosphoenolpyruvate carboxykinase does catalyzes the decarboxylation of oxaloacetate to phosphoenolpyruvate in the presence of guanosine triphosphate (GTP), but this is the different direction.

A carboxylation of a C3 into a C4 body is catalyses by phosphoenolpyruvate carboxylase in bacteria and e.g. C4 or CAM plants. In animals a carboxylation of a C3 into a C4 is realized by carboxylation of pyruvate to oxaloacetate by pyruvate carboxylase, which requires biotin and ATP ( e.g. during gluconeogenesis).

In my opinion, the mixing of animal, plant or bacterial enzymes by “other” authors (e.g. Salinas et al. or Chetri et al.) seems questionable. However, Blackstone & El Rahmany cited these published scientific works and refer to these data. Therefore, the most important question in this context is to substantiate that a carboxylation of phosphoenolpyruvate is indeed possible and to provide a convincing answer as to where the cytoplasmic oxaloacetates originate (in metazoans).

Could you give the free energy of the exact reaction of a C3 into a C4 body (via phosphoenolpyruvate carboxykinase (PEPCK)),  as well as the conditions (e.g. pH or CO2 concentration)?

It is the responsibility of the authors to critically examine the plausibility of the hypotheses of the underlying literature that you summarize in this manuscript and thus to correspond to the spirit of science, according to which well-founded, ordered knowledge that is considered to be reliable should be spread.

Minor revision:

The authors have written about the Warburg effect, but there is no citation of an original manuscript by Herr O. Warburg. I recommend the citing at least one original work by Warburg e.g. (Warburg, Posener, Negelein: Über den Stoffwechsel der Carcinomzelle. Biochemische Zeitschrift 1924, 152, 309–344).

Line 50-54 or line 109

The authors described:  “ …under long-term anaerobic conditions, cytosolic NADH is re-oxidized by, for example, forming malate from oxaloacetate. Malate can then be processed in mitochondria, with the additional NADH re-oxidized using a truncated electron transport chain, running the Krebs cycle backwards, and forming succinate, which may then be converted into products such as acetate and propionate before excretion.”

Question: Where is oxaloacetate produced under hypoxia and by what biochemical pathway (see also major revision)?  Is the malate-aspartate shuttle involved?  Please describe the second sentence and by what biochemical pathway do such pathways works in metazoans? These two sentences are very important for the authors’ argument and will be used later (line 109). This is because the malate shuttle does not normally function under anaerobic conditions as the oxidized form of the reducing equivalent is not sufficiently available. This could of course be different for cnidarians, nematodes, plathelminths, molluscs and annelids.

The authors wrote “ metazoans are not expected to operate the entire Krebs cycle in reverse, although many prokaryotes do this [13]. Line 55 

Question: Could these differences in the possible direction change of the TCA cycle be a problem to understand the Warburg effect in human cells compared to other metazoans?

Figure 1 visualizes this possible biochemical mechanism (see also major revision). However, this figure looks very similar to Figure 1 or 3 of the already published work Blackstone 2022, Oxygen 2, 493-502. Perhaps it would be possible to renew this figure 1 and distinguish it from the figures already published? The same applies to Figure 4 compared to Figure 4 Blackstone 2022, Oxygen 2, 493-502.

I am in no way suggesting that these illustrations are incorrect and do not fully describe the item. I'm just wondering if two very similar illustrations from an recent publication by the author that have been adapted to a new manuscript aren't also marked as such (e.g. adapted from...). This new reference would even make the visualization of a related work by the author possible and necessary.

The authors wrote: ” running the Krebs cycle backwards while using a portion of the electron transport chain…line 113 “metazoans are not expected to operate the entire  Krebs cycle in reverse, although many prokaryotes do this (line 55)”

Question: That is because RQ has so far only been described in nematodes, platyhelminths, mollusks and annelids (Tan et al., eLife 2020;9:e56376)? What does this mean for e.g. subphylum vertebrata, which do not express RQ, therefore running the Krebs cycle backwards is difficult? Moreover, in human cells, as Otto Warburg described this effect for these vertebrata-cells, malate appears to be a source of lactate production (DeBerardinis et al., PNAS USA 2007, 104, 19345-19350) but not a source of fumarate synthesis .

The authors wrote (line  165): “ that proliferative cells use, for example, malate dismutation would have” and and 167 “Malate then enters mitochondria without using the NADH shuttles” and “with greater ATP yield than lactate formation line” 168.

Question: Does this then mean that malate appears to be transported without a shuttle, even without the malate/aspartate shuttle. Can you provide a citation for such a "single" human cell malate shuttle? How many moles of ATP are produced under what circumstances and by what "type" of proton gradient (2 moles of ATP per mole of glucose by glycolysis and lactate redox product)? And again, where does the oxaloacetate come from?

In line 190 the authors wrote “ 2.3 One ring to rule them all:” .

That's a nice and witty aphorism for TCA, but if the major problems (major revision) are not answered in an acceptable manner, I must urge the authors to read not just Tolkien, but compelling papers showing, where all the crucial oxaloacetate, which is the basis of the hypothesis, actually originates. Well, the cited book “What is Life?” by Ewin Schrödinger was really a nice surprise. Thanks.

In general, Chapter 3 is very explanatory and appears to be a new  topic that differs from the previous arguments. Furthermore, RQ has only been described in nematodes, platyhelminths, mollusks and annelids (Tan et al., eLife 2020;9:e56376), therefore the analysis of UQ and RQ should focus on these affected phylum’s.

In Chapter 4, the authors wrote : “C. elegans exhibits anaerobic mitochondria, while H. sapiens does not” .

Question: What is the definition of anaerobic versus aerobic mitochondria? How important is the environment and what role do the four proteins actually play in your argumentation? Table 1 is not very meaningful. Rather should the authors place the four proteins one below the other as the amino acid sequence for all sea anemones, the worm and man?

In general, chapters 3-6 should focus more on cnidarian models and this argumentation could be expanded? I can hardly see the link to "An Organismic Perspective on the Warburg Effect" in these chapters, as too many other, few relevant facts are addressed.

Conclusion: The overall proliferation of a polyp is not really different from common metazoan models (e.g. worms, flies and mice) (line 429), which show limited cell proliferation. Indeed, cancer is unlike such organisms and even cnidarians because these cells show enormous and excessive proliferation. To what extent cnidarian models can depict this has not become plausible to me. Perhaps the manuscript would have developed more stringently if it had been limited to cnidarians and dealt exclusively with their metabolism.

Author Response

We thank the reviewer for the prompt return of their helpful review.  Below we respond to the comments that were made.

Comment: The manuscript submitted by Blackstone & El Rahmany : “An organismal perspective on the Warburg effect” conveyed the impression of an almost philosophical point of view on the effect described by Otto Warburg in the context of tumor, on the perspective of other organism and a general view. Nevertheless, a clearer focus of this work on Cnidarian, for which the authors have an indisputable expertise, would have given this review a distinct direction.

Response: In the revised manuscript, we have better articulated what our goals are.  Two robust fields of intellectual inquiry have existed independently for some time.  On the one hand, the Warburg effect is studied by biomedical scientists using human cells.  On the other hand, organismal biologists and physiologists investigate anaerobic metabolism using a host of non-model organisms.  Herein, we attempt to unite these distinct fields of inquiry.  It is not our goal to extend the boundaries of either field, nor to elaborate the rich detail of either.  Both fields have developed large literatures and active researchers.  Our goal is to connect these fields in the expectation that useful synergisms will result.  We realize that some specialists may not relish such connections and attempt to impose the standards of one field on the other.  This in our opinion would be missing the forest for the trees—and missing a rare opportunity.

Major Revision:

Comment: The author described a pathway and refers to it, which was described by different other authors e.g. | https://doi.org/10.1038/s41598-020-70202-y or https://doi.org/10.1016/j.bbabio.2020.148278.

The argumentation of these “other” authors is very similar:

Chetri et al.,: “During anaerobic metabolism, phosphoenolpyruvate (PEP) is converted to oxaloacetate via phosphoenolpyruvate carboxykinase (PEPCK), which is subsequently reduced by cMDH, reoxidizing the glycolytic NADH. This malate is transported to the mitochondria, where it is dismutated. Some malate is oxidized to acetate, while some are reduced to succinate and then metabolized to propionate.” https://doi.org/10.1038/s41598-020-70202-y

Salinas et al.,: “Glucose is converted into phosphoenolpyruvate by the glycolytic pathway, which is converted to oxaloacetate (not to pyruvate as in glycolysis), and then reduced to malate. Then, malate is imported into the mitochondrion, where a fraction is oxidized to acetate and CO2, and another fraction is converted to fumarate, which is then reduced to succinate by complex II.” “PEP is then converted to oxaloacetate (Oxac) by phosphoenolpyruvate carboxy-kinase” https://doi.org/10.1016/j.bbabio.2020.148278.

However, these interpretations of this pathway is the scientific basic for the argumentation of the manuscript submitted by Blackstone & El Rahmany .

I do not understand this pathway described by “other” authors (e.g. Salinas et al. or Chetri et al.), because in a biochemical sense phosphoenolpyruvate (PEP) cannot be converted into oxaloacetate. The reaction of a C3 into a C4-body is a carboxylation, not a convertion. Thus, carboxylation is not catalyzed by PEP, as stated by these “other” authors, because the phosphoenolpyruvate carboxykinase does catalyzes the decarboxylation of oxaloacetate to phosphoenolpyruvate in the presence of guanosine triphosphate (GTP), but this is the different direction.

A carboxylation of a C3 into a C4 body is catalyses by phosphoenolpyruvate carboxylase in bacteria and e.g. C4 or CAM plants. In animals a carboxylation of a C3 into a C4 is realized by carboxylation of pyruvate to oxaloacetate by pyruvate carboxylase, which requires biotin and ATP ( e.g. during gluconeogenesis).

In my opinion, the mixing of animal, plant or bacterial enzymes by “other” authors (e.g. Salinas et al. or Chetri et al.) seems questionable. However, Blackstone & El Rahmany cited these published scientific works and refer to these data. Therefore, the most important question in this context is to substantiate that a carboxylation of phosphoenolpyruvate is indeed possible and to provide a convincing answer as to where the cytoplasmic oxaloacetates originate (in metazoans).

Could you give the free energy of the exact reaction of a C3 into a C4 body (via phosphoenolpyruvate carboxykinase (PEPCK)),  as well as the conditions (e.g. pH or CO2 concentration)?

It is the responsibility of the authors to critically examine the plausibility of the hypotheses of the underlying literature that you summarize in this manuscript and thus to correspond to the spirit of science, according to which well-founded, ordered knowledge that is considered to be reliable should be spread.

Response: We are puzzled by this comment.  Our paper hinges on the ability of some eukaryotes, including metazoans, to carry out anaerobic metabolism without producing lactate.  In this context, we provide several references: two books and a lengthy review article, published over a 30-year period.  We realize of course that there are numerous other articles supporting these references, such as those pointed out by the reviewer.  Nevertheless, this is besides the main point of our article: such organisms can thus provide a valuable perspective on the Warburg effect, in which human cells carry out anaerobic metabolism in the presence of oxygen and produce lactate.  For our purposes, it does not matter what products are produced by these organisms or how they produce them—only that the products are not lactate.  If the details of malate dismutation were central to our manuscript, we would feel that a detailed response to the reviewer’s points would be necessary.  But this is not the case.  For the purposes of the manuscript, opine production would be just as relevant.  Question: Does the reviewer doubt for a moment that numerous metazoans produce succinate and related short-chain fatty acids under anaerobic conditions?  Studies too numerous to count have found this result.  Indeed, we now have preliminary mass spectrometry data showing that our hydroids do this under nutrient abundant conditions.

We focus on malate dismutation because this is well-documented by a large literature, with known genetic correlates.  We do not cite either of the references described by the reviewer, but from the quoted passages, they fit well within the extensive supporting literature.  Further, the description of this pathway in Martin et al. (2021) (which we do cite) seems to resolve the difficulties presented by the reviewer.  Their Figure 10 (p. 94), among others, clearly shows the conversion of phosphoenolpyruvate to oxaloacetate by the enzyme ATP-dependent phosphoenolpyruvate carboxykinase.  This reaction takes up carbon dioxide, and this would seem to resolve the stoichiometric issues raised by the reviewer.  We do not feel that a lengthy digression into the biochemical details of malate dismutation would benefit this manuscript, particularly in view of the lengthy, recently published treatise by Martin et al. that does exactly that.  Bill Martin is a good colleague, and we greatly respect his scholarship.

We suspect that the reviewer is unfamiliar with this organismal literature.  We encourage the reviewer to keep in mind that human biology may be a poor guide to some aspects of widespread eukaryotic and metazoan metabolic mechanisms, i.e., we suspect that the reviewer is suggesting that human cells cannot carry out the indicated metabolic reactions.  We do not doubt this.  Keep in mind that studies of many non-model organisms lack the funding and the tools that biologists studying human cells take for granted.  For example, the details of the RQ-based anaerobic metabolism were worked out only because of the possibility of therapeutic targets of human parasites.  We hope the reviewer will see the value of exposing a wide audience of biologists to the diversity of metazoan metabolism.

In the revised manuscript, we do provide some additional description of the information provided by Martin et al. to better direct readers to this valuable and comprehensive source.  We also point out that the conditions for these reactions are easy to describe.  Since these are marine organisms with limited homeostatic capabilities, the conditions are essentially those of seawater (pH 8.2, salinity roughly 30-35 parts per thousand, high levels of carbonates, calcium, and magnesium) and ambient temperatures.  In keeping with the reviewer’s comments, we also suggest that there may be aspects of cellular metabolism besides those directly related to RQ that also show functional differences in facultatively versus obligately aerobic mitochondria.

Minor revision:

Comment: The authors have written about the Warburg effect, but there is no citation of an original manuscript by Herr O. Warburg. I recommend the citing at least one original work by Warburg e.g. (Warburg, Posener, Negelein: Über den Stoffwechsel der Carcinomzelle. Biochemische Zeitschrift 1924, 152, 309–344).

Response: Good point.  We, however, have chosen to cite one of Warburg’s later papers (1956, Science), which is better known to the English-speaking audience.  Indeed, it is cited in Martin et al. (2021).  Further, this paper shows how Warburg maintained his hypothesis of respiratory impairment in cancer cells even after his data (Warburg, O. & Hiepler, E. 1952. Versuche mit Ascites Tumorzellen. Z. Naturforsch. 7b, 193–194) and those of others (Chance, B. & Castor, L. N. 1952. Some patterns of the respiratory pigments of ascites tumors in mice. Science 116, 200–202) indicated that this was not the case.

Comment: Line 50-54 or line 109

The authors described:  “ …under long-term anaerobic conditions, cytosolic NADH is re-oxidized by, for example, forming malate from oxaloacetate. Malate can then be processed in mitochondria, with the additional NADH re-oxidized using a truncated electron transport chain, running the Krebs cycle backwards, and forming succinate, which may then be converted into products such as acetate and propionate before excretion.”

Question: Where is oxaloacetate produced under hypoxia and by what biochemical pathway (see also major revision)?  Is the malate-aspartate shuttle involved?  Please describe the second sentence and by what biochemical pathway do such pathways works in metazoans? These two sentences are very important for the authors’ argument and will be used later (line 109). This is because the malate shuttle does not normally function under anaerobic conditions as the oxidized form of the reducing equivalent is not sufficiently available. This could of course be different for cnidarians, nematodes, plathelminths, molluscs and annelids.

Response: The reviewer raises an important point, which we will rephrase: if the malate-aspartate shuttle is used to move reducing equivalents (i.e., NADH) into mitochondria, there can be no net movement of malate.   This complicates claims that moving malate can side-step the NADH shuttles, although it can side-step one of the NADH shuttles (the glycerol 3-phosphate shuttle).  The last point by the reviewer is also a good one: human beings may be a poor guide for other metazoans, i.e., even though human beings are metazoans, they may not be representative.

As we outline above, Martin et al. (2021) generally describe in detail these pathways.  They are not explicit, however, on malate shuttles, and we can find little in the literature on this topic (we note this in the revised manuscript).  Martin et al. do include numerous metazoan examples, e.g., flatworms, roundworms, molluscs, annelids, and so on.  (Technically, of course human beings are metazoans too.)  They provide detailed descriptions and figures illustrating the relevant pathways, with numerous citations.  While individual taxa exhibit some idiosyncratic variation, the general themes of malate dismutation, for which we provide an overview, remain constant.  Martin et al. (2021) build on an earlier volume (Bryant 1991) and a lengthy review paper (Muller et al. 2012).  Thus, these data and concepts have been current in the literature for decades.  To date, of course, such data do not exist for any cnidarian that we are aware of.  Indeed, a major goal of this manuscript is to stimulate interest in exactly this problem.  Given the recent publication of the comprehensive Martin et al. volume, we do not feel that this manuscript should be a forum for further examination of the themes that Martin et al. so ably elaborate. 

In the revised manuscript, we point out that human biology may be a poor template for understanding other metazoans and eukaryotes.  Human cells exhibit highly derived characteristics and have likely lost characteristics that are shared by numerous other organisms.  Our elaboration of the RQ-based anaerobic metabolism is now presented in this context.  It may well be that there are other functional differences between human and other metazoan enzymes that have not yet been explored and could be related to amino acid sequence differences.  Nevertheless, we clarify the issues with regard to the movement of malate into mitochondria.

Again, malate dismutation is only one of several pathways that anaerobic metazoans might employ.  For the purposes of our manuscript, it does not matter which pathways are employed (e.g., it is known that cnidarians carry out opine formation); only that these pathways do not produce lactate.  However, we deem opine formation as unlikely to play a major role in proliferation and provide circumstantial genetic evidence that cnidarians are capable of an RQ-based anaerobic metabolism.  In the revised manuscript, we have further elaborated these issues.

Comment: The authors wrote “ metazoans are not expected to operate the entire Krebs cycle in reverse, although many prokaryotes do this [13]. Line 55 

Question: Could these differences in the possible direction change of the TCA cycle be a problem to understand the Warburg effect in human cells compared to other metazoans?

Response: With obligately aerobic mitochondria, human cells have very limited capabilities of anaerobic metabolism.  Indeed, as shown in our Table 1, in contrast to many metazoans, they lack even a vestige of RQ-based anaerobic metabolism.  Thus, we would slightly rephrase the question: could the lack of reversed TCA cycle and RQ-based metabolism in human cells be obscuring our understanding of the Warburg effect?  We would answer a resounding “yes.”  Indeed, this is exactly what we are suggesting in the manuscript.

Comment: Figure 1 visualizes this possible biochemical mechanism (see also major revision). However, this figure looks very similar to Figure 1 or 3 of the already published work Blackstone 2022, Oxygen 2, 493-502. Perhaps it would be possible to renew this figure 1 and distinguish it from the figures already published? The same applies to Figure 4 compared to Figure 4 Blackstone 2022, Oxygen 2, 493-502.

I am in no way suggesting that these illustrations are incorrect and do not fully describe the item. I'm just wondering if two very similar illustrations from an recent publication by the author that have been adapted to a new manuscript aren't also marked as such (e.g. adapted from...). This new reference would even make the visualization of a related work by the author possible and necessary.

Response: The reviewer’s point here is not clear.  Certainly, sketches of mitochondria and electron transport chains will bear necessary similarities.  Is it important to disguise such similarities?  In the revised manuscript, both figures have been revised.  In any event, the messages of these sketches here is very different from that of the Oxygen papers.

Comment: The authors wrote: ” running the Krebs cycle backwards while using a portion of the electron transport chain…line 113 “metazoans are not expected to operate the entire  Krebs cycle in reverse, although many prokaryotes do this (line 55)”

Question: That is because RQ has so far only been described in nematodes, platyhelminths, mollusks and annelids (Tan et al., eLife 2020;9:e56376)? What does this mean for e.g. subphylum vertebrata, which do not express RQ, therefore running the Krebs cycle backwards is difficult? Moreover, in human cells, as Otto Warburg described this effect for these vertebrata-cells, malate appears to be a source of lactate production (DeBerardinis et al., PNAS USA 2007, 104, 19345-19350) but not a source of fumarate synthesis .

Response: The focus of Tan et al. and related papers was developing targets to control human parasites, hence their limited exploration of metazoan diversity.  We expect that an RQ-based anaerobic metabolism will be widely found in marine intertidal animals, likely including even some basal chordates (e.g., intertidal ascidians).  Our genetic analysis strongly implies that Nematostella exhibits such a metabolism.  Human cells have completely lost such capabilities, so it is no surprise that malate is not processed in the same way.  In facultatively versus obligately aerobic mitochondria, likely there are other enzymatic differences related to malate dismutation.

Comment: The authors wrote (line 165): “ that proliferative cells use, for example, malate dismutation would have” and and 167 “Malate then enters mitochondria without using the NADH shuttles” and “with greater ATP yield than lactate formation line” 168.

Question: Does this then mean that malate appears to be transported without a shuttle, even without the malate/aspartate shuttle. Can you provide a citation for such a "single" human cell malate shuttle? How many moles of ATP are produced under what circumstances and by what "type" of proton gradient (2 moles of ATP per mole of glucose by glycolysis and lactate redox product)? And again, where does the oxaloacetate come from?

Response: As suggested above, we expect that the reviewer has identified another target (mitochondrial transporters) that can be investigated in the context of differences between facultatively and obligately aerobic mitochondria.  We expect that aspects of anaerobic metabolism other than those directly related to RQ also differ between human cells and metazoans that are capable anaerobically.  As noted in the revised manuscript, Martin et al. (2021) provide the details of the ATP yield of malate dismutation.  Malate dismutation adds about 3 moles of ATP to glycolysis, with about 1 mole from RQ-based phosphorylation.

Comment: In line 190 the authors wrote “ 2.3 One ring to rule them all:” .

That's a nice and witty aphorism for TCA, but if the major problems (major revision) are not answered in an acceptable manner, I must urge the authors to read not just Tolkien, but compelling papers showing, where all the crucial oxaloacetate, which is the basis of the hypothesis, actually originates. Well, the cited book “What is Life?” by Ewin Schrödinger was really a nice surprise. Thanks.

Response: Thank you.  Especially after Nick Lane’s wonderful description (Lane 2009 Life Ascending) of the TCA cycle (p, 25: “Down toward the bottom, somehow giving the impression that it is the centre of this insurrection of arrows, is a tight little circle, maybe the only circle, indeed the only ordered bit, on the whole map.  That’s it, the Krebs cycle.), we are surprised that we are the first to suggest the “one ring” parallel.  (Even Nick’s latest book, which was focused entirely on the Krebs cycle, did not say this.)  As for the oxaloacetate, as above we refer the reviewer to Martin et al.’s detailed book, and further note that human enzymes may be a poor guide to the capabilities of their homologues in other organisms.  (And we like Schrödinger’s book, too!)

Comment: In general, Chapter 3 is very explanatory and appears to be a new topic that differs from the previous arguments. Furthermore, RQ has only been described in nematodes, platyhelminths, mollusks and annelids (Tan et al., eLife 2020;9:e56376), therefore the analysis of UQ and RQ should focus on these affected phylum’s.

In Chapter 4, the authors wrote : “C. elegans exhibits anaerobic mitochondria, while H. sapiens does not” .

Question: What is the definition of anaerobic versus aerobic mitochondria? How important is the environment and what role do the four proteins actually play in your argumentation? Table 1 is not very meaningful. Rather should the authors place the four proteins one below the other as the amino acid sequence for all sea anemones, the worm and man?

Response: No doubt guided by the vagaries of funding, Tan et al. and related papers focused on human parasites.  Our goals are somewhat different; hence, we explore other taxa.  It will not come as a shock to the reviewer if we point out that some mitochondria only function in the presence of oxygen.  These are obligately aerobic.  Other mitochondria can function both in the presence of oxygen and in its absence.  These can be described as facultatively aerobic (or facultatively anaerobic).  We have clarified this in the revised manuscript.  It would make Table 1 unnecessarily cumbersome to include the entire sequences of these proteins, which of course are easily available in major databases.  We have, however, added footnotes to Table 1, listing the actual amino acids as suggested by Reviewer #2.

Comment: In general, chapters 3-6 should focus more on cnidarian models and this argumentation could be expanded? I can hardly see the link to "An Organismic Perspective on the Warburg Effect" in these chapters, as too many other, few relevant facts are addressed.

Response: Again, it is not our goal to present the totality of the field of metazoan anaerobic metabolism to the uninitiated.  With the recent publication of Martin et al.’s book, that would be foolish.  Nor do we present the Warburg effect in great detail.  Rather, our goal is to connect these heretofore independent fields of intellectual inquiry and to provide an entrée into the respective literatures.  We suspect that the reviewer has gained some perspective on the field of eukaryotic anaerobic metabolism.  This is our goal.  Since the Warburg effect is characterized by lactate secretion, doesn’t it make sense to examine metazoans that never secrete lactate when anaerobic?

Comment: Conclusion: The overall proliferation of a polyp is not really different from common metazoan models (e.g. worms, flies and mice) (line 429), which show limited cell proliferation. Indeed, cancer is unlike such organisms and even cnidarians because these cells show enormous and excessive proliferation. To what extent cnidarian models can depict this has not become plausible to me. Perhaps the manuscript would have developed more stringently if it had been limited to cnidarians and dealt exclusively with their metabolism.

Response: Cancer cells proliferate mitotically, but they never undergo meiosis.  Hence, it is not unreasonable to develop models of cancer that have the same characteristics.  Indeed, one of our cnidarian models (Eirene sp.) is the HeLa cell of hydroids.  It relentlessly overgrows and kills other encrusting species not by using the usual mechanisms of competition (hyperplastic stolons) but rather simply by smothering the competitor with its rapid and extreme proliferation.  In culture, it grows into large, ill-defined masses of tissue limited only by the available food.  We invite the reviewer to visit our lab and view this extraordinary creature.  (And yes, this is the one for which we have preliminary mass spec data showing short-chain fatty acid production under nutrient abundant conditions.)

Reviewer 2 Report

This review entiteld "An organismal perspective on the Warburg effect" suprisingly discous the Warburg effect from the other point of view. The Warburg effect is an specific metabolic mode which is presend in highly proliferating cells of human body, like stem cells but also cancer cells. An organismal perspective may help to clarify the biological nature of this effect where oxidative phosphorylation is reduced and lactic acid over-produced. 

The authors present perspectives of anaerobic metabolism in different organism with additional analysis of metabolic influence on proliferation. Common metazoan models (e.g. flys or mice) are not proper for this kind of evaluation. However, colonial marine hydroids with a metagenic life cycle can better fits here. Thus, nutrient-abundant and nutrient-scarce environment can be compared for physiological parameters related to the Warburg effect.

I find the text interesting with innovative point of view. In support, I have some additional suggestions:

1. In my opinion, the title can be enriched to cover the whole conception of review, e.g. An organismal perspective on the Warburg effect and models of proliferation studies. Of course, the authors can modify the title on the other way.

2. Figure 4: maybe it will be better to replace Q with RQ in (a) and Q with UQ in (b); the symbols will follow that used in the main text.

3. Table 1: full name of aa presented for four proteins can be described below the table.

Author Response

We thank the reviewer for the prompt return of their helpful review.  Below we respond to the comments that were made.

Comment: This review entitled "An organismal perspective on the Warburg effect" suprisingly discuss the Warburg effect from the other point of view. The Warburg effect is a specific metabolic mode which is present in highly proliferating cells of human body, like stem cells but also cancer cells. An organismal perspective may help to clarify the biological nature of this effect where oxidative phosphorylation is reduced and lactic acid over-produced.

Response: Thank you for the succinct summary.

Comment: The authors present perspectives of anaerobic metabolism in different organisms with additional analysis of metabolic influence on proliferation. Common metazoan models (e.g. flies or mice) are not proper for this kind of evaluation. However, colonial marine hydroids with a metagenic life cycle can better fits here. Thus, nutrient-abundant and nutrient-scarce environment can be compared for physiological parameters related to the Warburg effect.

Response: Again, thank you for the summary.

Comment: I find the text interesting with innovative point of view. In support, I have some additional suggestions:

Response: Thank you again.

Comment: 1. In my opinion, the title can be enriched to cover the whole conception of review, e.g. An organismal perspective on the Warburg effect and models of proliferation studies. Of course, the authors can modify the title on the other way.

Response: Good point. We have reflected on this point for some time and settled on: An organismal perspective on the Warburg effect and models for proliferation studies.

Comment: 2. Figure 4: maybe it will be better to replace Q with RQ in (a) and Q with UQ in (b); the symbols will follow that used in the main text.

Response: Yes, we agree that this will be clearer and have modified the figure accordingly, using R for rhodoquinone.

Comment: 3. Table 1: full name of aa presented for four proteins can be described below the table.

Response: Also, a good point.  We have added this information to the footnotes of the table.

Round 2

Reviewer 1 Report

Comment: The manuscript submitted by Blackstone & El Rahmany : “An organismal perspective on the Warburg effect”  

The authors gave many good answers to my questions about the first round of review. The most important concern, in my opinion, was the interpretations of this fundamental pathway published by “other  authors” (major concern). I’m not an expert on such specific biochemical pathways of other organisms, but I asked a professor of biochemistry about the reaction of PEPCK: His answer was: ”Typically, carboxylations are biotin- and ATP-dependent reactions. But there are also ATP-independent carboxylations. In this respect, I do not consider the complete reversibility of the PEPCK reaction to be implausible… the reverse reaction is dependent on a high CO2 concentration”.

Unfortunately, it is difficult to obtain information about books cited, e.g. B. Martin et al., because you have to buy such a book or use interlibrary loan, which takes some time to get the necessary information about the background of your manuscript. However, I found an interesting paper: (Latorre-Muro et al., 2018, Molecular Cell 71, 718-732 https://doi.org/10.1016/j.molcel.2018.07.031), which described “the conditions by which PCK1 triggers the synthesis of OAA” even for human cells. This might help convince conservative and old-fashioned readers that nature might be taking strange and unexpected ways, even in human cells. I would like to suggest citing this work in this manuscript.

So, I now assume that this "strange" biochemical pathway is possible. My biggest concern is therefore gone, although I still have doubts and some wonder. I therefore understand this manuscript as food for thought that could be published. In fact, we also wondered how nematodes organize their metabolism in a hypoxic atmosphere in the human colon. Thus, such a work helps to realize a networking of different areas.

 In 387 the authors wrote now:  “While this is clearly not the case (e.g., [41]),..” I think this fact is debated, but not fully understood.